# Understanding Label Bias in Single Positive Multi-Label Learning

**Julio Arroyo, Pietro Perona & Elijah Cole**
Caltech

## Abstract

Annotating data for multi-label classification is prohibitively expensive because every category of interest must be confirmed to be present or absent. Recent work on single positive multi-label (SPML) learning shows that it is possible to train effective multi-label classifiers using only one positive label per image. However, the standard benchmarks for SPML are derived from traditional multi-label classification datasets by retaining one positive label for each training example (chosen uniformly at random) and discarding all other labels. In realistic settings it is not likely that positive labels are chosen uniformly at random. This work introduces protocols for studying label bias in SPML and provides new empirical results.

## 1 Introduction

*Multi-label classification* aims to predict the presence or absence of every category from a collection of categories of interest (Tsoumakas & Katakis, 2007). This is a generalization of the standard *multi-class classification* setting, in which the categories are mutually exclusive. Multi-label classification is often more realistic than multi-class classification in domains like natural image classification where multiple categories tend to co-occur (Wang et al., 2016; Wei et al., 2015).

The primary obstacle to training multi-label classifiers is the expense of exhaustively annotating the presence and absence of every category of interest. To mitigate this problem, Cole et al. (2021) introduce the setting of *single positive multi-label* (SPML) classification, which aims to train effective multi-label image classifiers using only one positive label per image.

The SPML benchmarks introduced in Cole et al. (2021) are artificially generated from fully annotated multi-label classification datasets. For each training image, one positive label is chosen uniformly at random to be retained and all other labels are discarded. While this simple model is a reasonable starting point, it is unlikely to reflect realistic annotator behavior (Spain & Perona, 2011). If we want to apply SPML in realistic settings, we need to understand how different positive label selection biases affect the performance of SPML algorithms.

Our key contributions are as follows: (1) We extend the SPML benchmarks in Cole et al. (2021) by generating new label sets based on different models of annotator bias. (2) We provide the first study of the effect of label bias in SPML, filling a crucial gap in the literature with significant implications for the real-world applicability of SPML. The code and data are publicly available.[1]

## 2 Related Work

**SPML.** Cole et al. (2021) formalized the SPML setting and introduced benchmarks and algorithms for the problem. Subsequent work has used these benchmarks to develop new SPML algorithms (Kim et al., 2022; Abdelfattah et al., 2023; 2022; Ke et al., 2022; Zhou et al., 2022; Xu et al., 2022; Verelst et al., 2023). We complement this work by extending the benchmarks in Cole et al. (2021) and clarifying the role of label selection bias in SPML.

**Models of annotator behavior.** The benchmarks in Cole et al. (2021) are unlikely to reflect the behavior of real annotators, who are more likely to mention an object if it is larger or closer to the center of the frame (Berg et al., 2012) or may neglect to mention it if it is "too obvious" (Spain & Perona, 2011). We take inspiration from this literature for our models of annotator behavior.

---

[1] https://github.com/elijahcole/single-positive-multi-label/tree/spml-bias

## 3 BIAS MODELS FOR SINGLE POSITIVE MULTI-LABEL ANNOTATION

Let $(\mathbf{x}, \mathbf{y})$ denote a training example, where $\mathbf{x} \in \mathbb{R}^p$ is an image and $\mathbf{y} \in \{0, 1\}^L$ is a label vector. Suppose we ask an annotator to name one class in $\mathbf{x}$. We want to model the probability $P(i)$ that class $i \in \{1, \ldots, L\}$ will be selected. As in Cole et al. (2021), we assume $y_i = 0 \implies P(i) = 0$.

**Uniform bias.** Choose one positive label uniformly at random:

$$P_{\text{uniform}}(i) = \mathbb{1}_{[y_i=1]} \times 1/|\{i \in \{1, \ldots, L\} : y_i = 1\}|. \tag{1}$$

**Size bias.** Choose a category in proportion to the fraction of the image it occupies:

$$P_{\text{size}}(i) = \mathbb{1}_{[y_i=1]} \times A_i / \sum_{j=1}^{L} A_j \tag{2}$$

where $A_i$ is the the sum of the bounding box areas of all instances of class $i$.

**Location bias.** Choose a category more often if it occurs close to the center of the image:

$$P_{\text{location}}(i) = \mathbb{1}_{[y_i=1]} \times D_i^{-1} / \sum_{j=1}^{L} D_j^{-1} \tag{3}$$

where $D_i$ is the average distance from the center of all instances of class $i$.

**Semantic bias.** Determine probabilities based on empirical object spotting data:

$$P_{\text{semantic}}(i) = \mathbb{1}_{[y_i=1]} \times f_i / \sum_{j=1}^{L} \mathbb{1}_{[y_j=1]} f_j \tag{4}$$

where $f_i$ is the empirical spotting frequency for category $i$. See Appendix B for details.

## 4 RESULTS & DISCUSSION

All experiments are conducted on COCO (Lin et al., 2014). For each bias, we generate three independent realizations of SPML labels (see Appendix A.2). Then for each dataset realization and loss, we use the code from Cole et al. (2021) to train and evaluate SPML methods under the "Linear" protocol – see Cole et al. (2021) for implementation details. We present the results in Table 1. Additional bias implementation details in Appendix A.1.

**Table 1:** Test set mean average precision (MAP) results for different bias models and training losses (AN = "assume negative" loss, AN-LS = "assume negative" loss with label smoothing, ROLE = regularized online label estimation, EM = entropy maximization). Values for $P_{\text{uniform}}$ are copied from their respective papers.

| | $P_{\text{uniform}}$ **(1)** | $P_{\text{size}}$ **(2)** | $P_{\text{location}}$ **(3)** | $P_{\text{semantic}}$ **(4)** |
|---|---|---|---|---|
| $\mathcal{L}_{\text{AN}}$ (Cole et al., 2021) | 62.3 | $57.0 \pm 0.1$ | $61.0 \pm 0.2$ | $59.8 \pm 0.4$ |
| $\mathcal{L}_{\text{AN-LS}}$ (Cole et al., 2021) | 64.8 | $56.7 \pm 0.3$ | $62.7 \pm 0.1$ | $59.8 \pm 0.1$ |
| $\mathcal{L}_{\text{ROLE}}$ (Cole et al., 2021) | 66.3 | $60.1 \pm 0.1$ | $66.4 \pm 0.0$ | $\underline{66.4 \pm 0.0}$ |
| $\mathcal{L}_{\text{EM}}$ (Zhang et al., 2021) | $\underline{70.7}$ | $\underline{61.2 \pm 0.1}$ | $\underline{68.4 \pm 0.0}$ | $65.6 \pm 0.4$ |

We make three observations. (i) *The ranking of different methods is similar across bias types.* This provides some evidence that the standard $P_{\text{uniform}}$ benchmarks are a reasonable surrogate for more realistic biases that might be encountered in practice. (ii) *Performance often drops significantly when we change from $P_{\text{uniform}}$ to other biases.* Averaging over losses, the drop is largest for $P_{\text{size}}$ (-7.3 MAP) and smallest for $P_{\text{location}}$ (-1.4 MAP). (iii) *Different losses have different sensitivities to label bias.* $\mathcal{L}_{\text{AN-LS}}$ (-5.1 MAP) and $\mathcal{L}_{\text{EM}}$ (-5.7 MAP) lose more performance on average than $\mathcal{L}_{\text{AN}}$ (-3.1 MAP) and $\mathcal{L}_{\text{ROLE}}$ (-2.0 MAP).

These observations suggest that SPML algorithm rankings are roughly stable under different label biases, but the performance under $P_{\text{uniform}}$ tends to overestimate the performance under other biases. We hope that our work will facilitate future research on SPML under realistic conditions.

**Acknowledgements.** We thank Tsung-Yi Lin for providing the COCO object spotting data.

**URM Statement.** At least one key author meets the URM criteria of ICLR 2023 Tiny Papers Track.

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

## A  IMPLEMENTATION DETAILS

### A.1  BIAS MODELS

In this section we provide a few details for the non-trivial bias models: $P_{\text{size}}$, $P_{\text{location}}$, and $P_{\text{semantic}}$. The equations are reproduced here for convenience, but are identical to those in the main paper.

**Size bias.** We define

$$P_{\text{size}}(i) = \mathbb{1}_{[y_i=1]} \times A_i / \sum_{j=1}^{L} A_j$$

where $A_i$ is the the sum of the bounding box areas of all instances of class $i$. Note that we do not make any modifications to handle intersecting bounding boxes, so $\sum_{j=1}^{L} A_j$ may be larger than the image size.

**Location bias.** We define

$$P_{\text{location}}(i) = \mathbb{1}_{[y_i=1]} \times D_i^{-1} / \sum_{j=1}^{L} D_j^{-1}$$

where $D_i$ is the average distance from the center of all instances of class $i$. $D_i$ is implemented as the Euclidean distance between the center of the image and the center of the object bounding box. We also add a small value $\epsilon > 0$ to each $D_i$ to prevent division by zero for objects in the exact center of the image. $D_i^{-1}$ is defined to be zero if category $i$ is absent.

**Semantic bias.** We define

$$P_{\text{semantic}}(i) = \mathbb{1}_{[y_i=1]} \times f_i / \sum_{j=1}^{L} \mathbb{1}_{[y_j=1]} f_j$$

where $f_i$ is the empirical spotting frequency for category $i$, which is derived from annotator studies by the COCO team (Lin et al., 2014). Further details can be found in Appendix B.

### A.2  LABEL SET GENERATION

Like Cole et al. (2021), we generate SPML training data by sampling labels from a fully-labeled multi-label classification dataset as follows:

1. Choose a bias model $P \in \{P_{\text{uniform}}, P_{\text{size}}, P_{\text{location}}, P_{\text{semantic}}\}$.
2. Fix a random seed.
3. For each training image: (i) randomly choose one positive label according to $P$ and (ii) mark all other labels as "unknown".

For this work, we generate 3 realizations for each bias model with different seeds. Once generated, these SPML training sets are static, i.e. training labels are not re-sampled during training. The validation and test sets are always fully labeled.

## B  COCO OBJECT SPOTTING DATA

To build $P_{\text{semantic}}$, we use object spotting data collected according to the protocols in Lin et al. (2014). The data consists of annotations of the form (pixel_x, pixel_y, category_id) for images in the COCO 2014 official validation set. Note that this data was collected by the COCO authors in 2017, and is different from the original object spotting data used to prepare the COCO 2014 dataset.

We use this data to compute empirical object spotting frequencies $f_i$ for $i \in \{1, \dots, L\}$ as follows:

1. For each category, remove all spotting instances that do not fall inside of a bounding box of the same category.

2. Count the remaining spotting instances for category $i$ across all images, which we call $f_i$.

We then plug in the $f_i$ values to the definition of $P_{\text{semantic}}(i)$.

