# OpenReview forum: "Understanding Label Bias in Single Positive Multi-Label Learning"
_ICLR.cc/2023/TinyPapers — Submitted to Tiny Papers @ ICLR 2023_

### Official Review · Reviewer_ioca · 2023-03-31

**Confidence:** 3

**Summary Of Contributions:**

Introduce bias on single positive multi-label annotation without reproducible code

**Rating:**

Great Start (GS): a submission which meets some of the reviewing criteria but has room for improvement

**Strengths And Weaknesses:**

Strengths
1. paper introduces bias on SPML(single positive multi-label )annotation
2. table compares the performance on different label bias(uniform, size, location & semantic) & different loss

Weakness
1. the code is not provided where we are unable to determine if results are reproducible or the validity of the paper
2. Interpretation of the different loss subscripts mentioned in the table is missing

**Suggested Changes:**

1. Adding code helps the reviewers to determine the validity and verify the research is reproducible.
2. Below the table provide a expansion/meaning of the different losses with the subscripts : AN−LS , EM, AN, ROLE

---

### Official Review · Reviewer_i1h8 · 2023-04-04

**Confidence:** 3

**Summary Of Contributions:**

SPML uses a single positive label for each image, which is used to train the image classifier. At the same time, this method approaches the performance of traditional multilabel classifiers, its unlikely to give us behavior similar to that of our annotators.

**Rating:**

High Potential (HP): a submission which meets the reviewing criteria and has potential to make an impact on the field

**Strengths And Weaknesses:**


This work operationalizes certain notions of bias in the framework of SPML. The work appears to give evidence for P_uniform being a surrogate measure of bias.

Why does it matter that we emulate the behavior of that of our annotators? What happens if they are biased or the 'annotators' happen to be using biased models, i.e., ChatGPT, BERT models, etc.?


**Suggested Changes:**

Explain a bit more about why we wish to emulate behavior of our annotators perhaps.

I wonder if using multiple terms and which combination of multiple terms would help increase performance.

---

### Author Response · Authors · 2023-05-25
**Authors' response**

We thank the reviewers for their time and constructive feedback. We have uploaded a revised version that incorporates the following changes:
1. [ioca] The code to reproduce the results in Table 1 is now linked in the paper.
2. [i1h8, ioca] We have further edited the paper, clarified the language, and double checked that all notation is defined.

---

### Author Response · Authors · 2023-05-29
**Archival**

The authors wish to opt-in for archival.

---

### Meta-Review · Area_Chair_UXfY · 2023-04-06

**Recommendation:** Invite to archive
**Confidence:** 4

**Metareview:**

The paper explores the effect of label bias in single positive multi-label (SPML) learning. The authors argue that the standard benchmarks for SPML, which choose one positive label per image uniformly at random, may not be representative of real-world annotation settings. The paper introduces bias on SPML annotation and compares the performance of different label biases and loss functions.
The tackled problem of this paper seems to be noted by one reviewer as high potential.
Yet the other reviewer identified the lack of reproducible code and clear explanations for some terms as weaknesses.

**Summary:**

The paper examines label bias in single positive multi-label learning and its impact on standard benchmarks. Reviewers appreciated the novel analysis and comparisons, but noted the lack of reproducible code, unclear explanations, and insufficient discussion on annotator behavior and biases.

**Reason For Not Giving A Higher Recommendation:**

The concern raised by reviewers on reproducible codes and details needs to be addressed.

**Reason For Not Giving A Lower Recommendation:**

It seems that the problem setting in this paper is noted as has potential.

---

### Decision · Program_Chairs · 2023-04-08

Invite to archive